# A Patented Dietary Supplement (Hydroxy-Methyl-Butyrate, Carnosine, Magnesium, Butyrate, Lactoferrin) Is a Promising Therapeutic Target for Age-Related Sarcopenia through the Regulation of Gut Permeability: A Randomized Controlled Trial

**DOI:** 10.3390/nu16091369

**Published:** 2024-04-30

**Authors:** Mariangela Rondanelli, Clara Gasparri, Alessandro Cavioni, Claudia Sivieri, Gaetan Claude Barrile, Francesca Mansueto, Simone Perna

**Affiliations:** 1Department of Public Health, Experimental and Forensic Medicine, University of Pavia, 27100 Pavia, Italy; mariangela.rondanelli@unipv.it; 2Endocrinology and Nutrition Unit, Azienda di Servizi alla Persona ‘‘Istituto Santa Margherita’’, University of Pavia, 27100 Pavia, Italy; alessandro.cavioni01@universitadipavia.it (A.C.); claudia.sivieri01@universitadipavia.it (C.S.); gaetanclaude.barrile01@universitadipavia.it (G.C.B.); francesca.mansueto01@universitadipavia.it (F.M.); 3Division of Human Nutrition, Department of Food, Environmental and Nutritional Sciences (DeFENS), Università degli Studi di Milano, 20019 Milano, Italy; simoneperna@hotmail.it

**Keywords:** sarcopenia, hydroxy-methyl-butyrate, carnosine, magnesium, butyrate, lactoferrin, body composition, TNF-a, C-reactive protein, zonulin

## Abstract

Adequate diet, physical activity, and dietary supplementation with muscle-targeted food for special medical purposes (FSMP) or dietary supplement (DS) are currently considered fundamental pillars in sarcopenia treatment. The aim of this study is to evaluate the effectiveness of a DS (containing hydroxy-methyl-butyrate, carnosine, and magnesium, for its action on muscle function and protein synthesis and butyrate and lactoferrin for their contribution to the regulation of gut permeability and antioxidant/anti-inflammation activity) on muscle mass (assessed by dual X-ray absorptiometry (DXA)), muscle function (by handgrip test, chair test, short physical performance battery (SPPB) test, and walking speed test), inflammation (tumor necrosis factor-alpha (TNF-a), C-reactive protein (CRP), and visceral adipose tissue (VAT)) and gut axis (by zonulin). A total of 59 participants (age 79.7 ± 4.8 years, body mass index 20.99 ± 2.12 kg/m^2^) were enrolled and randomly assigned to intervention (*n* = 30) or placebo (*n* = 28). The skeletal muscle index (SMI) significantly improved in the supplemented group compared to the placebo one, +1.02 (CI 95%: −0.77; 1.26), *p* = 0.001; a significant reduction in VAT was observed in the intervention group, −70.91 g (−13.13; −4.70), *p* = 0.036. Regarding muscle function, all the tests significantly improved (*p* = 0.001) in the supplemented group compared to the placebo one. CRP, zonulin, and TNF-alpha significantly decreased (*p* = 0.001) in intervention, compared to placebo, −0.74 mg/dL (CI 95%: −1.30; −0.18), −0.30 ng/mL (CI 95%: −0.37; −0.23), −6.45 pg/mL (CI 95%: −8.71; −4.18), respectively. This DS improves muscle mass and function, and the gut muscle has emerged as a new intervention target for sarcopenia.

## 1. Introduction

Aging is often accompanied by the reduction and weakening of muscle mass, a condition defined as sarcopenia which today is formally recognized as a muscle disease with the diagnostic code ICD-10-MC [1]. According to the most recent criteria of the European Working Group on Sarcopenia in Older People (EWGSOP2), sarcopenia is considered probable when poor muscular strength is documented in the patient. The diagnosis of sarcopenia is then confirmed by the presence of poor quantity or quality of muscle mass; finally, it is considered severe when poor muscle strength, poor quantity and quality of mass, and poor physical performance are detected together [2].

According to a recent systematic review of 130 studies, sarcopenia is estimated to influence 10–16% of older people worldwide [3]. The prevalence of sarcopenia varied between studies and depending on the definition used. What is now known is that the prevalence of sarcopenia was higher among patients compared to general populations, and it varies according to the pathologies considered; for example, the prevalence ranges from 18% in diabetic patients to 66% in patients with unresectable esophageal cancer [3].

The onset of sarcopenia is favored by inflammation, immune-senescence, anabolic resistance, and increased oxidative stress. These mechanisms are further increased in cases of sedentary lifestyle, protein malnutrition, and comorbidities. Moreover, recently, there has been a growing interest from the scientific community regarding the role that gut microbiota could play in the pathophysiology of sarcopenia, hypothesizing the existence of a real gut–muscle axis [4]. In fact, advanced age not only affects the muscle but also causes gut microbiome dysbiosis and, therefore, alteration in gut permeability [5].

Fluctuations in the composition of the microbiota are frequent, especially in older people [6]. Over the age of 65, many changes occur in the intestinal microbiota in terms of biodiversity, composition, and functional characteristics [7]. Therefore, interventions via the gut–muscle axis may be a novel target to delay age-related muscle wasting and dysfunction [8].

Since an adequate diet, physical activity, and dietary supplementation are currently considered the fundamental pillars in the treatment and prevention of sarcopenia [4,9], developing a specific dietary supplement (DS) that slows the progression of sarcopenia is extremely important. Treating sarcopenia also means avoiding all the adverse consequences associated with it, including poor overall and disease-progression-free survival rate, postoperative complications, and longer hospitalization in patients with different medical situations as well as falls and fracture, metabolic disorders, cognitive impairment, and mortality in general populations [3].

The ideal muscle-targeted DS must contain various nutrients and bioactive molecules that have the capacity to modulate the multiple biological pathways that cause sarcopenia, such as inflammation, alteration in gut permeability, anabolic resistance, nutritional (in particular, amino acids) deficiencies, and increased oxidative stress. 

Given this background, the aim of this study is to evaluate the effectiveness of a muscle-targeted DS that contains multiple substances with different actions in order to counterbalance the loss of lean mass in association with a personalized diet and adequate physical activity in elderly sarcopenic subjects. Specifically, the muscle-targeted DS contains hydroxy-methyl-butyrate (HMB), carnosine, and magnesium for their direct action on muscle function and on protein synthesis, and butyrate and lactoferrin for their contribution to the regulation of gut permeability and for antioxidant/anti-inflammation activity.

## 2. Materials and Methods

### 2.1. Population

A randomized, double-blind, placebo-controlled trial was conducted in sarcopenic patients aged 55–85 years old. Patients randomly received treatment or a placebo. The subjects were recruited from the Dietetic and Metabolic Unit of the “Santa Margherita” Institute, University of Pavia, Italy.

We used the revised definition of EWGSOP2 to define sarcopenia in men as low handgrip strength (HGS) ≤27 kg if men, ≤15 if women, or five times chair—stand tests (5-STS) as >15 s. Subjects who were 65 years of age or older and with a body mass index (BMI) of 20 to 30 kg/m^2^ were recruited. Participants with severe kidney illness, moderate-to-severe hepatic failure, endocrine disorders, psychiatric disorders, cancer (within the last five years), or hypersensitivity to any provided FSMP component were not allowed to participate in the trial. The recruitment of the subjects for the study is shown in Figure 1.

### 2.2. Standard Protocol Approval, Registration, and Patient Consent

This study was approved by the Ethics Committee of the University of Pavia, Italy (approval number 0112/29072022) and complied with the ethical standards as laid down in the 1964 Declaration of Helsinki, with written informed consent obtained from all patients entering the pre-treatment phase. This study was registered under ClinicalTrials.gov (accessed on 19 September 2022) (NCT05730985).

The presence of adverse events was reported by subjects as well as open-ended inquiries by members of the research staff.

### 2.3. Study Design and Sample Size

This was a randomized (1:1), double-blind, placebo-controlled, parallel-group, 4-month clinical intervention study. The study was conducted at the Geriatric Physical Medicine and Rehabilitation Division of Santa Margherita Hospital, Azienda di Servizi alla Persona, Pavia, Italy. Participants were allocated to the intervention groups via a computer-generated random blocks randomization list, and random assignments were concealed in sealed envelopes.

The sample size calculation was based on the reference study of Rondanelli et al. [10], and the sample size on the primary outcome, as an increase in lean mass of 0.6 kg in the placebo group and +1.6 kg in the intervention group (+1 kg increase in lean mass in the intervention compared to placebo) as a percentage (+3.97% treated group and +1.51% placebo) “between groups” (+2.5% increase in lean mass) on treatment variable for 4 months of treatment (continuous variable). A sample size of 60 patients (30 patients per arm) would be needed for enrollment, according to estimates that took into account two balanced groups with 1:1 ratio allocation (n1 = n2), an effect size of 0.5, an alpha significance level set at 0.05, a dropout rate of 10%, and 80% power in detecting differences between groups. The experimental design timeline is shown in Figure 2.

### 2.4. Nutritional Assessment and Interventions

A customized food plan was provided to every participant. For each patient in both groups, a customized nutritional plan was created, including 1.5 grams of protein per kilogram of body weight each day. A qualified dietitian calculated the estimated basal metabolic rate by multiplying the predicted activity factor by the Harris–Benedict formula, which was then used to calculate calorie intake. The recommended diet called for a cautious intake of roughly 55% carbs and 30% fats. During the course of four months, compliance with the dietary intervention was evaluated using a 24-h dietary recall once per month.

Subjects were randomly allocated to receive the experimental formula twice daily: calcium hydroxymethylbutyrate 1500 mg, l-carnosine 125 mg, Lactoferrin 50 mg, Sodium butyrate 250 mg, Magnesium 150 mg (Italian patent n. 102017000050442, pending European patent, EP3634458, pending PCT n. WO2018207122, pending Chinese patent n. CN110913882) or the control formula: isocaloric placebo with the same flavor. The experimental and the control formulas were delivered as an indistinguishable water-dispersible powder.

The compliance of subjects with an intake of DS was monitored by inputting the number of daily servings consumed in a diary.

### 2.5. Biochemical Parameters

Plasma samples were analyzed using enzyme-linked immunosorbent assay kits for zonulin (Cat # K5601, Immundiagnostik AG, Bensheim, Germany). The CRP level was measured by particle-enhanced immunonephelometry on a Behring Nephelometer analyzer using the relevant kit (Dade Behring, Marburg, Germany). Serum levels of TNF-α were assessed using the TNF-α Enzyme-Linked Immunosorbent Assay (ELISA) test (Biosource International, Human TNF-α, Vilvoorde, Belgium).

### 2.6. Anthropometric Measurements

Anthropometric measurements were assessed at the beginning of the study (at baseline; (t0) and after 4 months (t2). Body weight and height were measured following a standardized technique [11], and BMI was then calculated. The weight was measured using a mechanical tilting scale (Wunder Sa.bi Srl Mod C 201); the subjects wore only underwear. The height was measured through the use of the same instruments, including a statimeter composed of a graduated vertical bar and a movable branch perpendicular to the bar and on this slide, which is placed on the highest point of the head. The subject is measured barefoot. Both instruments used are calibrated monthly. All the anthropometric parameters were measured by the same investigator.

### 2.7. Body Composition Assessment 

Body composition represented by fat-free mass (FFM) and fat mass (FM) was measured using a Lunar Prodigy DXA (GE Medical Systems, Fairfield, CT, USA). The in vivo coefficients of variation were 0.89% for the whole FM and 0.48% for FFM. Visceral adipose tissue (VAT) volume was estimated using a constant correction factor of 0.94 g/cm^3^. This software automatically places a quadrilateral box that represents the android region outlined by the iliac crest, with a superior height of 20% of the distance from the top of the iliac crest to the base of the skull [12]. The calculation of the appendicular lean mass (ALM) was based on the sum of the fat-free mass from the arms and legs. ALM was standardized by BMI according to previous studies [13,14] in order to normalize the data by height and weight. Measurements were performed at baseline (t0) and after 4 months (t1).

### 2.8. Muscle Strength Evaluation

The hydraulic hand dynamometer (Jamar 5030 J1, Sammons Preston Rolyan, Bolingbrook, IL, USA) was used to assess the handgrip strength of the muscles in accordance with established methods, and the accuracy was 0.6 N. With the elbow by the side of the body and the arm at right angles, the subject applies an isometric contraction while holding the dynamometer in the hand that will be examined. Muscle strength was assessed at times t0 and t1. 

### 2.9. Physical Performance Assessment

Physical performance was assessed using the (a) SPPB, which comprised gait speed [15]), and balance (three different tests that assess the ability to stand with the feet together in the side-by-side, semi-tandem, and tandem positions); each component was scored from 0 (not possible) to 4 (best performance). This assessment was performed at t0 and t2.

Every patient had individualized training in aerobic and endurance exercises. All patients were placed on an individual, moderate-level (Borg Rate of Perceived Exertion scale score of 12–14) physical fitness and muscle mass-promoting program [16]. Every patient's unique ability to exercise was tracked throughout each session under the supervision of qualified personnel, and the intensity level was modified as needed. Five days a week of daily exercise sessions comprised the physical examination. The length of the first sessions was twenty minutes, and it could go up to 30 minutes, depending on how hard the exercises were. The following was covered in each session: a five-minute warm-up, followed by a progressive sequence of five to ten minutes of seated to standing muscle-strengthening exercises: heels, toes, knee lifts, and knee extensions (while seated); hip flexions and lateral leg raises (while standing next to a chair for stability); ankle-weight bearing exercises, using weights that range from 0.5 to 1.5 kg based on each participant’s strength as resistance increased gradually; hip and leg extensions (using resistance bands). There were also upper-body exercises, which included bicep curls and double-arm pulldowns. After that, patients were instructed to repeat the exercises eight times if necessary. The exercises included five to ten minutes of balance and gait exercises, such as tandem stands, one-leg stands, multidirectional weight shifts, and tandem walks, as well as practicing proper gait mechanics, which included increasing stride length and maintaining balance while changing directions and/or patterns of gait. The session concluded with a five-minute cool-down. Once the length of each exercise session was stabilized at 30 min, and no increase in exercise intensity could be undertaken for five consecutive days, a multidisciplinary team consisting of a geriatrician, physiatrist, physiotherapist, and nurse made the decision to conclude rehabilitation and release the patient.

### 2.10. Statistical Analysis

Statistical analysis and the reporting of this study were conducted in accordance with the Consolidated Standards of Reporting Trials (CONSORT) guidelines, with the primary analysis based on the full analysis set. The normal distribution of the variables was checked using the Kolmogorov-Smirnov test and using Q-Q graphs. Descriptive statistics representing raw data for each group (control and treatment) and the full sample were provided, including mean, standard deviation, and frequencies, where appropriate. After verification of the normal distribution of the continuous variables, data were analyzed and statistically compared intra and between groups using repeated measures and adjusting by age and gender. Variances were defined as statistically significant for *p*-value < 0.05. All analyses were performed using the Statistical Package for the Social Sciences, version 28.0 (SPSS Inc., Chicago, IL, USA). 

## 3. Results

The study included a total of 59 adult sarcopenic patients (16 males and 43 females; 11 males and 19 females in the intervention group, and 5 males and 24 females in the placebo group); the data were collected from September 2022 to November 2023. The characteristics of the patients at baseline are shown in Table 1. No significant differences between the groups at baseline were observed.

Table 2 shows the mean difference changes in the primary and secondary outcomes. The changes differ significantly between the groups (supplement minus placebo effect) for all the variables considered.

Statistically significant changes in anthropometric parameters and the body composition variable were recorded; body weight and BMI increase in the intervention group versus the placebo, +3.47 kg (CI 95%: −3.02; 3.92) and +1.27 kg/m^2^ (CI 95%: −0.83; 1.71), respectively (*p* = 0.001). The SMI significantly improved in the supplemented group compared to the placebo group, +1.02 (CI 95%: −0.77; 1.26), *p* = 0.000; moreover, a significant reduction in VAT was observed in the intervention group, −70.91 g (CI 95%: −137.13; −4.70), *p* < 0.036.

When considering muscle function, all the tests (handgrip test, chair test, SPPB test, and walking speed test) significantly improved (*p* < 0.001) in the supplemented group compared to placebo, 8.92 kg (CI 95%: −6.68; 10.87), −7.40 s (CI 95%: −8.93; −5.84), 2.96 (CI 95%: −2.26; 3.64), −0.33 s (CI 95%: −0.42; −0.24), respectively.

Lastly, CRP, zonulin, and TNF-alfa significantly decreased (*p* = 0.000) in the intervention group, compared to placebo, −0.74 mg/dL (CI 95%: −1.30; −0.18), −0.30 ng/mL (CI 95%: −0.37; −0.23), −6.45 pg/mL (CI 95%: −8.71; −4.18), respectively.

Figure 3 shows the Pearson’s R heatmap correlations between the mean difference chances of selected biomarkers in the treatment group. Dark blue color represents a strong positive correlation, while dark red color represents a negative correlation.

An inflammatory biomarker, such as a TNF-α decrease, was directly correlated with a decrease in the level of zonulin (r = 0.35). The SMI increment was associated with a positive increase in the handgrip test (r = 0.33) and SPPB (r = 0.81).

## 4. Discussion

There are two main results of this study: 1. the DS-induced improvement of skeletal muscle and functional performance in elderly sarcopenic patients; 2. the presence of a gut–muscle axis in elderly patients with sarcopenia.

These results are due to the capacity of the different components of DS to modulate multiple biological pathways and acknowledged major properties helpful for the health of muscle mass: hydroxy-methyl-butyrate (HMB), carnosine, magnesium, for their direct action on muscle function and on protein synthesis, and butyrate and lactoferrin for their contribution to the regulation of gut permeability and for antioxidant/anti-inflammation activity. So, this DS fulfills the need for multifactorial action potential that is useful for counterbalancing a loss of muscle mass and function.

Firstly, the DS administered to the patients contains HMB, carnosine, and magnesium, known to exert direct action on muscle function and protein synthesis.

HMB is a metabolite of leucine and has proven useful in the attenuation of loss of mass, strength, and muscle function. The older diseased population might benefit from dietary HMB because of its established positive properties and its long-lasting pharmacological effect. The pharmacokinetics of leucine are characterized by a very fast absorption and distribution, while HMB is characterized by slow metabolism in the body. It, therefore, has a more prolonged effect on muscle protein synthesis and breakdown rates [17]. A review including 203 patients aged ≥60 years with sarcopenia and frailty revealed an increase in lean mass, while strength and muscle function were preserved thanks to HMB supplementation [18]. Another study demonstrated that HMB supplementation certainly helps preserve muscle mass during a 10-day period of bed rest in healthy older adults [19]. HMB was useful in improving the increase in mass and strength of skeletal muscles in subjects who perform physical activity and in older people, while further studies are needed to better examine the effects of HMB administered alone [20].

Carnosine is a natural bioactive endogenous dipeptide, composed of β-alanine and l-histidine, present at 99% in skeletal muscle [21]. The deterioration of the human organism in old age may be associated with reduced tissue concentrations of carnosine and, therefore, a lack of its antioxidant capacities [22]. Sarcopenia can be counteracted by improving systemic carnosine reserves through beta-alanine (β-alanine) supplementation [23]. A very recent review reported that an adequate intake of carnosine (up to 2 g per day) can slow the process of sarcopenia, aging, and the development of age-related diseases [24].

Concerning magnesium, the benefits of its consumption have been demonstrated in elderly subjects regarding the increase in muscle mass [25], physical performance, and, therefore, the reduction of the prevalence of sarcopenia [26]. In a cross-sectional study conducted by the EWGSOP2, 2532 on elderly participants, 1310 men and 1222 women, divided into sarcopenic and non-sarcopenic groups, it was found that sufficient oral magnesium intake was beneficial in preventing muscle loss [27]. Moreover, a randomized controlled trial conducted on healthy older women found that the intake of 300 mg of magnesium per day led to a significantly greater improvement in physical performance, as assessed by Short Physical Performance Battery (SPPB), chair test, and 4-minute walking test, compared to the control group, suggesting magnesium plays a role in preventing decline in physical performance and sarcopenia [28].

Secondly, the DS tested in this trial contains butyrate and lactoferrin, known for their contribution to the regulation of gut permeability and antioxidant/anti-inflammation activity.

The benefits brought by butyrate are related to its inhibiting action on histone deacetylase. To date, there are few human studies that have evaluated the association between gut microbiota and muscle mass. Among them, an observational study of 23 subjects in a nursing home found that patients with higher frailty scores (assessed by the Rockwood Clinical Frailty Scale) had a lower representation of butyrate-producing bacteria in their microbiota (in particular Clostridium XIVa and Ruminococcus) [29]. An Italian study suggested the role of two other SCFA-producing species, Faecalibacterium Prausnitzii and Roseburia, in the pathogenesis of sarcopenia in a sample of older people [30]. A 2021 Chinese study evaluated the causal association between gut butyrate synthesis and skeletal muscle mass in a sample of postmenopausal women. The results of this study also confirm the role of the SCFA-producing microbiota in the maintenance of muscle mass [31].

Skeletal muscle and the development of sarcopenia are influenced by the gut microbiota, which in turn is influenced by the intake of whey protein [32]. The digestive process of whey protein contributes to the formation of potent antimicrobial peptides derived from whey, such as lactoferrin, a basic glycoprotein with pleiotropic impact on the activation of the innate immune system, which directly suppresses inflammation via NfKB-dependent mechanisms, especially in the gastrointestinal tract [33,34]. A preliminary study of the oral administration of bovine lactoferrin in postmenopausal women demonstrated significant reductions in CRP and IL-6 levels [35], showing its positive activity in counterbalancing inflammation. A substantial modification of the microbiota during aging may influence changes in intestinal physiology, such as reduced motility, reduced mucus, barrier dysfunction, and dysbiosis, which may mediate bacterial toxin translocation and muscle aging.

This study confirmed the hypothesis of the existence of a gut–muscle axis in elderly patients with sarcopenia and, consequently, the importance of the components included in the tested DS.

The reduction in plasma zonulin levels following treatment, in fact, indicates the repair of the intestinal mucosal barrier, as zonulin is a critical regulator of intestinal permeability [36]. A recent study reported an association of plasma zonulin with the sarcopenia phenotype in the older population [37]. Moreover, it has also been shown that in sarcopenic patients with chronic obstructive pulmonary disease (COPD), zonulin, a marker of intestinal permeability, is a predictor of sarcopenia [38,39]. Butyrate exerts protective effects by enhancing intestinal barrier function and activating the FFA2 receptor-mediated PI3K/Akt/mTOR pathway [39]. Butyrate plays a vital role in mucosal repair by non-transglutaminase-mediated as well as transglutaminase-mediated pathways, dilatation of arterioles, increase in oxygen uptake and mucosal blood flow, reduction of mucosal permeability as well as increase in mucosal production and its release [40]. Therefore, recent advances elucidated that interventions via the muscle-gut axis have the potential to reverse the sarcopenia phenotype [41], and our study confirms the effectiveness of such an intervention. In addition to the positive action on intestinal permeability, another interesting result of the study is the reduction of the inflammatory state, as demonstrated by the statistically significant reduction of inflammatory markers such as TNF-a, CRP, and VAT. Chronic inflammation can affect catabolic processes in skeletal muscles and lead to a decline in physical performance [42]. Visceral adipose tissue is linked to a number of inflammatory mechanisms [43]. Visceral adipose tissue, compared to subcutaneous adipose tissue, produces a greater quantity of inflammatory markers, a greater production of pro-inflammatory and prothrombotic adipokines, and is characterized by greater lipolytic activity, with greater release of free fatty acids (FFA) into the circulation [43]. The extent of the macrophage infiltrates correlates directly with the grade of adiposity and with the state of systemic inflammation. The inflammatory infiltrate is caused by different mechanisms: (1) macrophages accumulate around necrotic adipocytes in adipose tissue; so, the pro-inflammatory state is due, at least in part, to a reduced ability of the macrophage to “clean” adipose tissue from apoptotic cells. (2) excessive lipolysis or an excessive influx of FFA exposes the adipocyte to an excessive load of fatty acids; FFA can bind the TLR4 complex, activating the inflammatory response. (3) Adipocyte necrosis induced by ischemia leads to recruitment of macrophages [44].

Elevated TNFα in aged muscle is associated with decreased muscle force production [45,46]. TNFα is also linked to sarcopenia because this pro-inflammatory cytokine is known to be associated with other factors that contribute to sarcopenia, including protein degradation, reactive oxygen species (ROS) accumulation, and apoptosis [47,48]. In addition, TNFα may be associated with sarcopenia by promoting insulin resistance, delaying muscle repair, and exacerbating the pro-inflammatory response by up-regulating IL-6 [49,50,51]. Pro-inflammatory cytokines such as TNFα, which have been found to be elevated in aging skeletal muscle, also promote ROS accumulation [52].

As far as CRP and inflammation are concerned, in a meta-analysis by Bano et al., sarcopenia was reported to be connected with higher CRP values [53], and in a recent study, Zupo et al. found that the predicted risk factors for the development of sarcopenia seemed to be CRP, according to Random Forest selection [54].

In conclusion, considering that sarcopenia is a pathology with multifactorial causes, such as inflammation, alteration in gut permeability, anabolic resistance, nutritional (in particular amino acids) deficiencies, and increased oxidative stress, a multitarget therapy that counterbalances these causes is necessary. The DS studied in this randomized clinical trial had these multitarget purposes, which were achieved, as demonstrated by the results. We hypothesize that the association between hydroxy-methyl-butyrate, carnosine, and magnesium allowed a statistically significant increase in muscle mass (SMI increase) and an improvement in muscle performance (increase in handgrip and SPPB and reduction in chair-to-stand test), while the presence of butyrate and lactoferrin reduced inflammation (as demonstrated by the reduction of CRP, TNF-a, and VAT) and contributed to the regulation of intestinal permeability (as demonstrated by the reduction of zonulin), thus also demonstrating that the gut–muscle axis has emerged as a new intervention target for sarcopenia.

The correlations found are also interesting. The TNF-α decrease was directly correlated with a decrease in the level of zonulin, suggesting a link between inflammation and gut permeability. The SMI increase was associated with a positive increase in the handgrip test and SPPB, demonstrating a simultaneous increase in mass, strength, and performance. This result is very interesting because it confirms that this dietary supplement acts on the three aspects that concern muscle: mass, strength, and performance.

This study has certain limitations. We did not investigate the participants’ amount and quality of diet, which can affect the gut microbiome. Moreover, gut microbial synthesis of the SCFA, serum butyrate levels, and other cytokines (e.g., interleukin 6) could have been investigated. In any case, nobody abandoned the study due to taste and odor considered pleasant through an evaluation obtained from an internal survey. Moreover, another important limitation concerns the fact that it is not possible to identify the extent to which each nutrient can be inferred to have contributed to the results. A further limitation of the study concerns the greater number of women compared to the number of men, but to test the effect times of treatments (between groups) and within-group treatments, we applied univariate models adjusting for gender (and age), including as a covariate in order to overcome this gender difference.

The dietary supplement was well tolerated, and there were no serious adverse events. Compliance was 100%.

Thus, additional research is required, in particular, to delineate the relationship and contribution of the gut–muscle axis to sarcopenia.

## 5. Conclusions

The findings of this study indicate that the administration of the patented DS based on HMB, carnosine, magnesium, lactoferrin, and butyrate (Sitrophin patent) appears to be an emerging valid strategy to counteract the progression of sarcopenia and sarcopenia-defining parameters in older adults.

## Figures and Tables

**Figure 1 nutrients-16-01369-f001:**
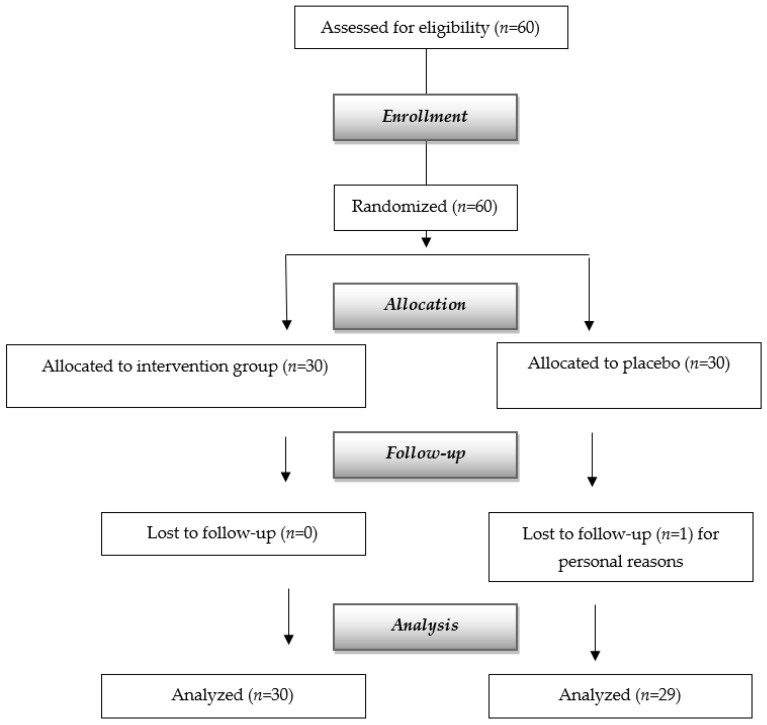
Flow diagram of the study.

**Figure 2 nutrients-16-01369-f002:**
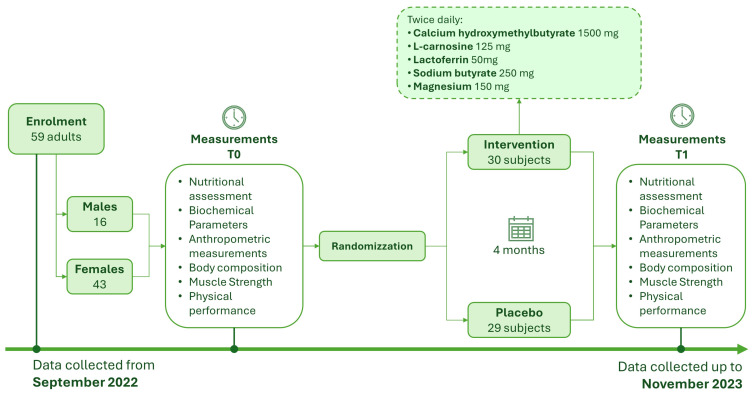
Timeline of the experimental design.

**Figure 3 nutrients-16-01369-f003:**
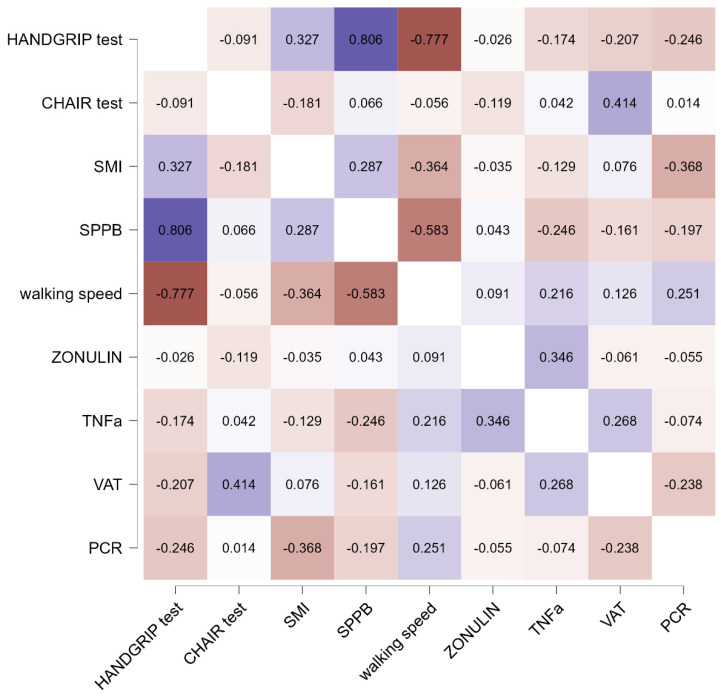
Pearson’s r Heatmap.

**Table 1 nutrients-16-01369-t001:** The baseline characteristics of patients between the intervention and placebo groups.

Variable	Placebo Group (*n* = 29)	Intervention Group (*n* = 30)	Total (*n* = 59)	*p*-Value between Groups
Diseases (*n*)	4.17 ± 1.37	4.10 ± 1.13	4.14 ± 1.24	0.825
Medicines (*n*)	4.10 ± 1.24	4.00 ± 1.02	4.05 ± 1.12	0.726
Height (m)	1.55 ± 0.08	1.59 ± 0.09	1.57 ± 0.09	0.055
Weight (kg)	51.59 ± 8.84	53.26 ± 8.35	52.44 ± 8.56	0.459
BMI (kg/m^2^)	21.12 ± 2.21	20.87 ± 2.06	20.99 ± 2.12	0.651
Handgrip test (kg)	17.41 ± 4.30	17.27 ± 3.41	17.34 ± 3.84	0.885
Chair test (s)	24.52 ± 5.50	25.11 ± 6.73	24.82 ± 6.11	0.717
SPPB test (score)	4.48 ± 0.51	4.63 ± 1.10	4.56 ± 0.86	0.504
Walking speed (m/s)	1.92 ± 0.28	1.81 ± 0.32	1.86 ± 0.30	0.164
SMI	5.36 ± 0.42	5.50 ± 0.57	5.43 ± 0.50	0.277
Fat Mass (kg)	15.78 ± 6.92	14.66 ± 5.59	15.21 ± 6.25	0.498
VAT (g)	757.61 ± 236.94	739.13 ± 378.61	748.21 ± 314.42	0.824
CRP (mg/dL)	1.43 ± 1.47	0.99 ± 1.33	1.21 ± 1.41	0.228
Zonulin (ng/mL)	2.80 ± 0.22	2.82 ± 0.22	2.81 ± 0.22	0.686
TNF-alfa (pg/mL)	12.87 ± 4.81	12.68 ± 4.50	12.78 ± 4.61	0.877

Abbreviations: *n*, number; BMI, body mass index; SPPB, short physical performance battery; SMI, skeletal muscle index; VAT, visceral adipose tissue; CRP, c-reactive protein.

**Table 2 nutrients-16-01369-t002:** Within-group (pre-post) and between-group (treatment minus placebo effect) mean difference changes from baseline (from day 0 to the end of the supplementation) for all investigated variables during the supplementation period.

Variable	Placebo Group (*n* = 29); Intra-Group Δ Change (CI 95%)	Intervention Group (*n* = 30); Intra-Group Δ Change (CI 95%)	Effects between Groups (Intervention Minus Placebo) Δ Change (CI 95%)	*p*-Value between Groups *
Weight (kg)	−1.31 (−1.62; −0.99)	2.13 (1.85; 2.47)	3.47 (3.02; 3.92)	**0.001**
BMI (kg/m^2^)	−0.37 (−0.67; −0.06)	0.91 (0.60; 1.21)	1.27 (0.83; 1.71)	**0.001**
Handgrip test (kg)	−2.76 (−4.13; −1.39)	6.17 (4.82; 7.51)	8.92 (6.68; 10.87)	**0.001**
Chair test (s)	1.99 (0.91; 3.06)	−5.41 (−6.47; −4.35)	−7.40 (−8.93; −5.84)	**0.001**
SPPB test (score)	0.38 (0.10; 0.86)	3.33 (2.86; 3.81)	2.96 (2.26; 3.64)	**0.001**
Walking speed (m/s)	0.11 (0.05; 0.17)	−0.22 (−0.28; −0.16)	−0.33 (−0.42; −0.24)	**0.001**
SMI	−0.22 (−0.39; −0.05)	0.80 (0.63; 0.97)	1.02 (0.77; 1.26)	**0.001**
Fat Mass (kg)	−1.24 (−2.39; −0.822)	1.55 (0.42; 2.69)	2.79 (1.15; 4.43)	**0.001**
VAT (g)	−4.53 (−51.01; 41.94)	−75.45 (−121.12; −29.78)	−70.91 (−137.13; −4.70)	**0.036**
CRP (mg/dL)	0.24 (−0.16; 0.63)	−0.50 (−0.89; −0.12)	−0.74 (−1.30; −0.18)	**0.011**
Zonulin (ng/mL)	0.02 (−0.03; −0.07)	−0.28 (−0.33; −0.23)	−0.30 (−0.37; −0.23)	**0.001**
TNF-alfa (pg/mL)	0.85 (−0.74; 2.44)	−5.60 (−7.16; −4.04)	−6.45 (−8.71; −4.18)	**0.001**

* in bold: value with *p* < 0.05. Abbreviations: *n*, number; BMI, body mass index; SPPB, short physical performance battery; SMI, skeletal muscle index; VAT, visceral adipose tissue; CRP, c-reactive protein.

## Data Availability

The original contributions presented in the study are included in the article; further inquiries can be directed to the corresponding author.

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
