# Peer review of "A Patented Dietary Supplement (Hydroxy-Methyl-Butyrate, Carnosine, Magnesium, Butyrate, Lactoferrin) Is a Promising Therapeutic Target for Age-Related Sarcopenia through the Regulation of Gut Permeability: A Randomized Controlled Trial"

_nutrients, 2024, doi:10.3390/nu16091369_

Round 1
Reviewer 1 Report
Comments and Suggestions for Authors
I accept the data as significant data that leads to improvement of sarcopenia through continuous intake of multiple nutrients including HMB. To clarify the significance of this study, we would like to see improvements in the following areas
(1) The appropriateness of the male/female ratio at the time of grouping (especially, the difference between the groups was observed in 11 and 5 males. I expect the validity of this result to be reinforced by using stratified analysis data for females only, etc.) In addition, a discussion of gender differences is also needed.
(2) In Nutrients 2022 14, 4566, in addition to pre- and post-trial data, body composition data at the midpoint of the study are also available.
(3) I think a discussion of the supplements taken, including the extent to which each nutrient can be inferred to have contributed to these results, is needed.
The following is a detailed point.
Is the total value in Table 1 Handgrip test correct?
I think the unit of "Fat Mass" should be kg.
The unit of CRP is listed only up to mg/d.
Table 2: Shouldn't the mean change from baseline be listed and the data in % be omitted?
I think you should clarify the findings from this study regarding Fig. 2.
Author Response
Dear Reviewer,
Thank you for your suggestions.
The changes in the text are yellow highlighted.
The paragraphs written in red have been modified to reduce the degree of similarity as required.
Thank you for you cooperation,
The authors
REV 1
I accept the data as significant data that leads to improvement of sarcopenia through continuous intake of multiple nutrients including HMB. To clarify the significance of this study, we would like to see improvements in the following areas
-
The appropriateness of the male/female ratio at the time of grouping (especially, the difference between the groups was observed in 11 and 5 males. I expect the validity of this result to be reinforced by using stratified analysis data for females only, etc.) In addition, a discussion of gender differences is also needed.
ANSWER:In order to test the effect time for treatment (between grpups) and intra group, we applied the univariate model adjusting by gender (along with age) that was included as covariate. Furthermore, the covariate gender was not statistically significant in the model. For this reason, we did not stratify the results by gender.
-
In Nutrients 2022 14, 4566, in addition to pre- and post-trial data, body composition data at the midpoint of the study are also available.
ANSWER: considering the different times of taking the supplements in the 2 studies, it was decided to carry out only the per-post evaluation and not the evaluation at the intermediate time.
-
I think a discussion of the supplements taken, including the extent to which each nutrient can be inferred to have contributed to these results, is needed.
ANSWER: in the discussion new sentences have been added on the topic suggested, particularly in the section regarding the limitations of the study.
The following is a detailed point.
Is the total value in Table 1 Handgrip test correct?
ANSWER: the placebo value entered was incorrect. It has been modified.
I think the unit of "Fat Mass" should be kg.
ANSWER: the unit has been changed to kg.
The unit of CRP is listed only up to mg/d.
ANSWER: the unit has been correct as “mg/dl”
Table 2: Shouldn't the mean change from baseline be listed and the data in % be omitted?
ANSWER: the column with % has been deleted.
I think you should clarify the findings from this study regarding Fig. 2.
ANSWER: new sentences have been added in the discussion on the results shown in figure 2.
Reviewer 2 Report
Comments and Suggestions for Authors
This study investigates the effects of a DS containing ingredients such as hydroxy-methyl-butyrate, carnosine, magnesium, butyrate, and lactoferrin, aimed at addressing sarcopenia. The study enrolled 59 participants, divided into two groups: one supplemented with the DS and the the control placebo group. The study assessed muscle mass, conducted various muscle function tests and walking speed tests, and measured markers of inflammation (TNF-a, CRP) as well as gut barrier function (zonulin, as a marker of gut permeability). The results suggest that the DS not only effectively increases muscle mass and improves muscle function but also has beneficial effects as an anti-inflammatory agent and increases gut barrier health. The study underscores the gut-muscle axis as a novel target for sarcopenia intervention, highlighting the potential of specific nutritional supplements in the treatment and management of sarcopenia, thereby improving the quality of life for older adults. The manuscript is well-written and interesting.
A few suggestions to improve the manuscript are made here:
1. The introduction should provide a comprehensive background and outline the current state of knowledge on the manuscript's topic. I miss information regarding the epidemiology of sarcopenia and the importance of such research.
2. A full characterization of DS should be included to provide comprehensive insights into its properties and functionalities.
3. The manuscript would benefit from including a timeline of the experimental design. This addition would clarify the methodology employed by the authors, aiding in the understanding of the experimental design and subsequent parts of the study.
4. Comprehensive description of the reagents and equipment, including the names of producers, cities, states (if applicable in the US), and countries should be provide (in some analysis missing). Including these details would significantly enhance the manuscript's clarity and reliability.
5. The description of the anthropometric methods provided in the manuscript is very brief. A more detailed explanation is needed. This should include specifics on how measurements were taken, the equipment used, and any calibration processes to ensure accuracy.
6. Considering that the safety assessment section contains only one sentence, it may not necessitate a separate paragraph. Integrating this information into a related section could streamline the manuscript and improve the flow of information.
7. Figure 1, which primarily illustrates aspects of the study's methodology, would be more suitably placed in the Methodology section. Relocating Figure 1 would offer clearer insight into the study’s procedures.
8. In Table 1, the abbreviation (n) is used, which might not be immediately clear to all readers. It is presumed to stand for 'number,' commonly referring to the number of subjects or samples within each group. To enhance clarity and prevent any potential confusion, it would be beneficial to explicitly define this abbreviation the first time it is used or consider replacing it with a more descriptive term.
9. The legend of Figure 2 should include the explanation of the circle size. Additionally, the numbers within the figure are somewhat blurry and could benefit from clarification regarding their significance. It would also be advantageous to increase the font size of the measurement names to improve readability.
10. The manuscript presents interesting outcomes on the improved gut barrier permeability due to the DS. It would be beneficial to include a few sentences explaining the potential molecular mechanisms underlying these improvements.
11. In line 332, the authors mention that visceral adipose tissue is associated with various inflammatory mechanisms. For a more comprehensive understanding, could the authors please list these specific mechanisms?
Author Response
Dear Reviewer,
Thank you for your suggestions.
The changes in the text are yellow highlighted.
The paragraphs written in red have been modified to reduce the degree of similarity as required.
Thank you for you cooperation,
The authors
REV 2
This study investigates the effects of a DS containing ingredients such as hydroxy-methyl-butyrate, carnosine, magnesium, butyrate, and lactoferrin, aimed at addressing sarcopenia. The study enrolled 59 participants, divided into two groups: one supplemented with the DS and the control placebo group. The study assessed muscle mass, conducted various muscle function tests and walking speed tests, and measured markers of inflammation (TNF-a, CRP) as well as gut barrier function (zonulin, as a marker of gut permeability). The results suggest that the DS not only effectively increases muscle mass and improves muscle function but also has beneficial effects as an anti-inflammatory agent and increases gut barrier health. The study underscores the gut-muscle axis as a novel target for sarcopenia intervention, highlighting the potential of specific nutritional supplements in the treatment and management of sarcopenia, thereby improving the quality of life for older adults. The manuscript is well-written and interesting.
A few suggestions to improve the manuscript are made here:
-
The introduction should provide a comprehensive background and outline the current state of knowledge on the manuscript's topic. I miss information regarding the epidemiology of sarcopenia and the importance of such research.
ANSWER: New sentences have been added on epidemiology of sarcopenia and on the relevance of the study on the therapy of sarcopenia
-
A full characterization of DS should be included to provide comprehensive insights into its properties and functionalities.
ANSWER: A full characterization of DS was presented in the discussion.
The manuscript would benefit from including a timeline of the experimental design. This addition would clarify the methodology employed by the authors, aiding in the understanding of the experimental design and subsequent parts of the study.
ANSWER: a new figure has been added with timeline of the experimental design.
-
Comprehensive description of the reagents and equipment, including the names of producers, cities, states (if applicable in the US), and countries should be provide (in some analysis missing). Including these details would significantly enhance the manuscript's clarity and reliability.
ANSWER: the materials and methods section was revised in consideration of the reviewer's requests
-
The description of the anthropometric methods provided in the manuscript is very brief. A more detailed explanation is needed. This should include specifics on how measurements were taken, the equipment used, and any calibration processes to ensure accuracy.
ANSWER This section has been implemented.
-
Considering that the safety assessment section contains only one sentence, it may not necessitate a separate paragraph. Integrating this information into a related section could streamline the manuscript and improve the flow of information.
ANSWER: this paragraph has been deleted and the sentence has been inserted in paragraph 2.2
-
Figure 1, which primarily illustrates aspects of the study's methodology, would be more suitably placed in the Methodology section. Relocating Figure 1 would offer clearer insight into the study’s procedures.
ANSWER: figure 1 has been relocated in Methodology section
8. In Table 1, the abbreviation (n) is used, which might not be immediately clear to all readers. It is presumed to stand for 'number,' commonly referring to the number of subjects or samples within each group. To enhance clarity and prevent any potential confusion, it would be beneficial to explicitly define this abbreviation the first time it is used or consider replacing it with a more descriptive term.
ANSWER: the abbreviation (n) has been clarified
9. The legend of Figure 2 should include the explanation of the circle size. Additionally, the numbers within the figure are somewhat blurry and could benefit from clarification regarding their significance. It would also be advantageous to increase the font size of the measurement names to improve readability.
ANSWER: the figure has been replaced with a better one. An explanatory sentence has been added.
10. The manuscript presents interesting outcomes on the improved gut barrier permeability due to the DS. It would be beneficial to include a few sentences explaining the potential molecular mechanisms underlying these improvements.
ANSWER: new sentences have been added.
11. In line 332, the authors mention that visceral adipose tissue is associated with various inflammatory mechanisms. For a more comprehensive understanding, could the authors please list these specific mechanisms
ANSWER: the paragraph has been implemented.
Round 2
Reviewer 1 Report
Comments and Suggestions for Authors
I think it's a unit of. "Fat mass" should be in kg. Answer: Change the unit to kg. ⇒ could not confirm. Are there any errors in the fat mass data listed in Table 2?
I think it should be made clear the findings of this study regarding Figure 2. Answer: New text has been added to the discussion of the results shown in Figure 2. ⇒Confirmed.
1. Validity of the male-female ratio at the time Grouping (in particular, the difference between the groups was 11 and 5 males. The validity of this result is based on stratified analysis data for females only). It is also necessary to discuss gender differences. ANSWER: To test the effect times of treatments (between GRPUPs) and within-group treatments, we applied univariate models adjusting for gender (and age). included as a covariate. Additionally, the covariate gender is statistically significant in the model. For this reason, we did not stratify the results in the following way. gender. ⇒I understand the reason for not conducting stratified analysis. However, I think that discussion comments are necessary due to gender differences in reference values such as grip strength, and the impact on women as reported by Kim et al.
Nutrients 2022 14, 4566, In addition to pre- and post-trial data, midpoint body composition data of the study is also available. Answer: Considering that the two studies took supplements at different times, we only performed post-by-post evaluations and did not perform mid-time assessments. ⇒ I understand.
3. I think it is necessary that the supplements taken (including the extent to which each nutrient can be inferred) contributed to these results. Answer: With a new discussion text on the proposed topic has been added, especially in the section About the limitations of the study. ⇒Confirmed. I understand that the discussion of each nutrient, but no comment on why it was prescribed, was adopted, especially regarding its significance or superiority. A mixture containing a volume.
Author Response
REVIEWER 1, REPORT 2
I think it's a unit of. "Fat mass" should be in kg. Answer: Change the unit to kg. ⇒ could not confirm. Are there any errors in the fat mass data listed in Table 2?
ANSWER: we apologize for the mistake. The fat mass unit has been changed to kg.
I think it should be made clear the findings of this study regarding Figure 2. Answer: New text has been added to the discussion of the results shown in Figure 2. ⇒Confirmed.
-
Validity of the male-female ratio at the time Grouping (in particular, the difference between the groups was 11 and 5 males. The validity of this result is based on stratified analysis data for females only). It is also necessary to discuss gender differences. ANSWER: To test the effect times of treatments (between GRPUPs) and within-group treatments, we applied univariate models adjusting for gender (and age). included as a covariate. Additionally, the covariate gender is statistically significant in the model. For this reason, we did not stratify the results in the following way. gender. ⇒I understand the reason for not conducting stratified analysis. However, I think that discussion comments are necessary due to gender differences in reference values such as grip strength, and the impact on women as reported by Kim et al.
ANSWER: in the part regarding the limitations of the study we added a sentence regarding sex differences, explaining how, through statistical analysis, we overcame this problem.
Nutrients 2022 14, 4566, In addition to pre- and post-trial data, midpoint body composition data of the study is also available. Answer: Considering that the two studies took supplements at different times, we only performed post-by-post evaluations and did not perform mid-time assessments. ⇒ I understand.
3. I think it is necessary that the supplements taken (including the extent to which each nutrient can be inferred) contributed to these results. Answer: With a new discussion text on the proposed topic has been added, especially in the section About the limitations of the study. ⇒Confirmed. I understand that the discussion of each nutrient, but no comment on why it was prescribed, was adopted, especially regarding its significance or superiority. A mixture containing a volume.
Reviewer 2 Report
Comments and Suggestions for Authors
The manuscript has been greatly improved through the revision process, I recommend adding an explanatory sentence to the legend of Figure 3 to clarify the meaning of the circle size. However, I didn't see this addition in the revised version of the manuscript, contrary to what was stated in the authors' response.
Comments on the Quality of English Language
There are awkward phrases in newly written parts. Please check his sentences.
The manuscript has been greatly improved through the revision process, and I support its publication after the suggested minor corrections are addressed. I recommend adding an explanatory sentence to the legend of Figure 3 to clarify the meaning of the circle size. However, I didn't see this addition in the revised version of the manuscript, contrary to what was stated in the authors' response.
Author Response
REVIEWER 2, REPORT 2
Comments and Suggestions for Authors
The manuscript has been greatly improved through the revision process, I recommend adding an explanatory sentence to the legend of Figure 3 to clarify the meaning of the circle size. However, I didn't see this addition in the revised version of the manuscript, contrary to what was stated in the authors' response.
ANSWER: We replaced the circle heatmap with a numbered heatmap with correlation coefficients in high definition
Comments on the Quality of English Language
There are awkward phrases in newly written parts. Please check his sentences.
ANSWER: the English language has been reviewed. We attach the certification.
